# Sequential modification of bacterial chemoreceptors is key for achieving both accurate adaptation and high gain

Bernardo A. Mello [1,2], Anderson B. Beserra[2] & Yuhai Tu [1✉]

Many regulatory and signaling proteins have multiple modification sites. In bacterial chemotaxis, each chemoreceptor has multiple methylation sites that are responsible for adaptation. However, whether the ordering of the multisite methylation process affects adaptation remains unclear. Furthermore, the benefit of having multiple modification sites is also unclear. Here, we show that sequentially ordered methylation/demethylation is critical for perfect adaptation; adaptation accuracy decreases as randomness in the multisite methylation process increases. A tradeoff between adaptation accuracy and response gain is discovered. We find that this accuracy-gain tradeoff is lifted significantly by having more methylation sites, but only when the multisite modification process is sequential. Our study suggests that having multiple modification sites and a sequential modification process constitute a general strategy to achieve both accurate adaptation and high response gain simultaneously. Our theory agrees with existing data and predictions are made to help identify the molecular mechanism underlying ordered covalent modifications.

[1] IBM T. J. Watson Research Center, Yorktown Heights, New York, NY 10598, USA. [2] Physics Institute, University of Brasilia, Brasilia 70919-970, Brazil. ✉email: yuhai@us.ibm.com

Most post-translational regulatory processes involve reversible covalent modifications (phosphorylation/ dephosphorylation, methylation/demethylation, etc.) of key proteins catalyzed by enzymes (kinase/phosphatase, methyl-tranferase/methylesterase, etc.). Instead of having only a single site of modification, many regulatory proteins such as histones, p53, RNA polymerase II, tubulin, etc. have multiple modification sites[1]. The multiple modification sites allow a single regulatory protein to have complex functions depending on combinations of different modification processes[2]. For example, the histone proteins have multiple covalent modification sites of different types (methylation, acetylation, phsophorylation, etc.) and the different combinations of the multiple modification sites are thought to code for different gene expression patterns in different cells[3]. However, how this combinatorial molecular code works, i.e., how it is encoded and decoded, remains poorly understood[4].

One well-studied multisite regulatory protein is the cyclin-dependent kinase inhibitor Sic1 in *Saccharomyces cerevisiae* (yeast). Sic1 has more than six phosphorylation sites whose main function is regulating the timing of the G1/S transition in yeast cell cycle[5,6]. Huang and Ferrell[7] first suggested that the response sensitivity can be enhanced by having multiple modification sites. However, Gunawardena[8] pointed out that other effects such as substantial disparities in enzyme efficiency among different sites are also needed in making a sharp switch. Later work by Salazar and Hofer[9] showed that a random phosphorylation process among the different sites gives rise to a shallow but rapid response while sequential processing gives rise to a steeper but slower response. Though much progress have been made, dynamics and functions of multisite modification in Sic1 remain not fully understood.

In this paper, we focus on a relatively simpler signaling system, bacterial chemotaxis[10], where multisite modification has an important role in adaptation[11]. Adaptation is an important general biological behavior that allows a living system to adjust its internal state in response to changes in its environment so that it can return to a set activity level after a fast response to a persistent change in the external stimulus[12]. In bacterial chemotaxis, a chemoreceptor has multiple methylation sites. The kinase activity of a chemoreceptor is determined by chemoeffector ligand concentration (external stimulus) as well as the receptor methylation level (internal state)—a higher attractant concentration leads to a lower kinase activity and a higher methylation level leads to a higher kinase activity. Adaptation in bacterial chemotaxis is achieved by a feedback mechanism in which the receptor methylation level (internal state of the receptor) is controlled by a methyltransferase CheR and a methylesterase CheB that act at a time scale much longer than the response time to a change in external stimulus (ligand concentration). The catalytic efficiencies of CheR and CheB depend on the receptor activity, which form the feedback mechanism for adaptation[13–16]. However, despite the general consensus on the importance of a negative feedback mechanism for accurate adaptation in bacterial chemotaxis, the detailed receptor methylation/demethylation kinetics among the multiple methylation sites remain unclear.

How do different methylation kinetics, random or sequential, affect adaptation accuracy and response gain? What are the benefits to have multiple modification sites? In this paper, we address these questions by systematically investigating effects of different multisite modification processes, from purely random to strictly sequential, on system-level functions such as adaptation accuracy and signal amplification. Our theoretical findings allow us to infer the multisite modification dynamics from existing experimental data. More importantly, our study leads to specific suggestions of future experiments to determine the molecular mechanism controlling the multisite modification dynamics.

## Results

**Modeling multisite modification dynamics.** In previous modeling studies of the receptor methylation (demethylation) reactions, the microscopic methylation state of the receptor ($\mu$) has been ignored. Here, we consider the transitions between the $2^M = 16$ ($M = 4$ is the total number of modification sites) individual microscopic methylation states of a receptor explicitly. As shown in Fig. 1, all the states $\mu$ are grouped (column-wise) by their total methylation level $m = \sum_{j=1}^{4} \mu_j$, so a methylation (demethylation) reaction moves the current state to another state in the column to the right (left). However, among the multiple states in the next column which one does it transition to? And at what rate? Here, we consider two cases, one special and one general, as shown in Fig. 1a, b, respectively.

For the special case of strictly sequential modification, which is implicitly assumed in previous models[17,18], the methylation and demethylation processes follow the same sequence (in opposite directions) among the five states shown in Fig. 1a. Following previous work[13–15], the negative feedback control is implemented by only allowing methylation (demethylation) of the inactive (active) receptors respectively. If we define $k_m^+$ and $k_m^-$ as the

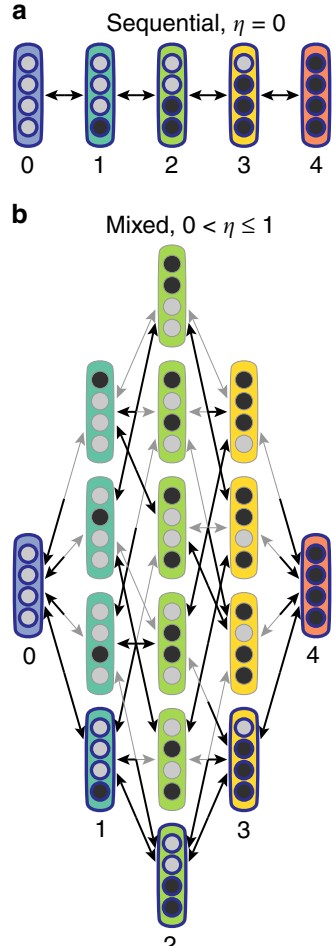

**Fig. 1 Illustration of the methylation/demethylation dynamics of a receptor with *M* = 4 modification sites. a** The purely sequential case $\eta = 0$. Methylated and unmethylated sites are labeled by filled and hollow circles, respectively. The total modification level $m$ is shown below each group (column) of receptor modification states with the same $m$. **b** The mixed case with $0 < \eta \leq 1$. The sequential modification reactions are represented by the black arrows. The non-sequential transitions are represented by the gray arrows when $0 < \eta \leq 1$.

average methylation and demethylation rates for all receptors with the same total methylation level $m$, this negative feedback mechanism leads to:

$$k_m^+ = k^+(1 - \langle a \rangle_m) , \quad k_m^- = k^- \langle a \rangle_m , \quad (1)$$

where $\langle a \rangle_m$ is the average activity of receptors with methylation level $m$ and the kinetic rates, $k^+$ and $k^-$, are proportional to CheR and CheB concentrations, respectively.

In the general case, when site $j-1$ is methylated ($\mu_{j-1} = 1$), the methylation rate for the next site in the sequence $j$ in state-$\mu$ is given by the same sequential methylation rate $k_m^+$ as in the strictly sequential case described above. However, when site $j-1$ is not methylated ($\mu_{j-1} = 0$), methylation of site $j$ can still occur via the random methylation process albeit with a smaller rate $k_m^{+\mathcal{R}} = \eta k_m^+$ where $\eta$ is a parameter ($0 \le \eta \le 1$) characterizing the randomness in the methylation process. Combining these two possibilities, the site-specific methylation rate $\tilde{k}_j^+$ for site $j$ can be written as:

$$\tilde{k}_j^+ = k_m^+[\mu_{j-1} + \eta(1 - \mu_{j-1})] . \quad (2)$$

Similarly, demethylation of site $j$ depends on whether site $j+1$ is demethylated, and the site-specific demethylation rate $\tilde{k}_j^-$ for site $j$ can be written as:

$$\tilde{k}_j^- = k_m^-[(1 - \mu_{j+1}) + \eta \mu_{j+1}] , \quad (3)$$

where $k_m^\pm$ in Eqs. (2) and (3) are the sequential methylation and demethylation rates given by Eq. (1). To describe modifications of the boundary states, i.e., the fully unmethylated state ($m = 0$) and the fully methylated state ($m = M$), we introduce a forward initiator with $\mu_0 = 1$ for methylation of the $j = 1$ site in Eq. (2) and a reverse initiator with $\mu_{M+1} = 0$ for demethylation of the $j = M$ site in Eq. (3).

Despite the same feedback mechanism given by Eq. (1), dynamics of receptor methylation level depends on the degree of randomness ($\eta$) in the multisite modification process. In this paper, we investigate consequences of different multisite modification schemes, from sequential to random, by studying behaviors of the standard model of bacterial chemotaxis for different values of $0 \le \eta \le 1$. In particular, we study how the adaptation error $\xi$ and the response gain $\Gamma$ are affected by $\eta$. Details of the full standard model framework for studying bacterial chemotaxis and the precise definition of $\xi$ and $\Gamma$ are given in the Methods section.

**Sequential modification is essential for perfect adaptation.** In general, the adapted activity level $\langle a \rangle^{\mathcal{A}}([L])$ is a function of the ligand concentration $[L]$. Adaptation is deemed perfect if $\langle a \rangle^{\mathcal{A}}$ is a constant independent of $[L]$. For a general case $0 < \eta < 1$, we can determine the adapted activity by solving the full model numerically using Monte-Carlo method (see Supplementary Methods for details). However, simple analytical equations for $\langle m \rangle$ can be found for the extreme cases $\eta = 0$ and $\eta = 1$, which provide insight on a key condition for perfect adaptation:

$$\frac{d\langle m \rangle}{dt} = k^+(1 - \langle a \rangle) - k^- \langle a \rangle + \epsilon , \quad \text{(when } \eta = 0\text{)} \quad (4)$$

$$\frac{d\langle m \rangle}{dt} = k^+ \langle (M - m)(1 - \langle a \rangle_m) \rangle - k^- \langle m \langle a \rangle_m \rangle , \quad \text{(when } \eta = 1\text{)} \quad (5)$$

where $\langle a \rangle$ is the average receptor activity and the term $\epsilon$ comes from the boundary effects at $m = 0$ and $m = M$ (see Supplementary Note 1 for details of the derivation).

For the case of purely sequential methylation ($\eta = 0$), the right hand side of Eq. (4) has the remarkable property of only explicitly

depending on $\langle a \rangle$ but not on $\langle m \rangle$ or $[L]$. As a result, the adapted activity $\langle a \rangle^{\mathcal{A}} \approx k^+/(k^+ + k^-)$ is independent of $[L]$, i.e., perfect adaptation[17,18]. The independence of the methylation rate $\frac{d\langle m \rangle}{dt}$ on $m$ is that only one modification site is available for methylation or demethylation per receptor at any given time when modification reactions are sequential.

Figure 2a illustrates the adaptation process in response to a series of step increase in ligand concentration $[L]_1 \to [L]_2 \to [L]_3$. The solid line represents the adapted activity obtained by setting the right hand side of Eq. (4) to zero. The dashed curves represent the activity as a function of $\langle m \rangle$ for different values of $[L]$. Upon a sudden increase of $[L]$, say from $[L]_1$ to $[L]_2$, the system first responds by decreasing its activity as represented by the downward arrow (blue) as illustrated in Fig. 2a. This altered activity triggers the adaptation mechanism that slowly increases $m$, causing the system to follow the upward arrow (green) along the dashed line for $[L]_2$ until it reaches the adapted activity level that is roughly independent of $[L]$. The fundamental reason for perfect adaptation is that $\langle a \rangle^{\mathcal{A}}$ is independent of $\langle m \rangle$, i.e., the solid line in Fig. 2a is flat for a large range of $\langle m \rangle$.

For the case of random methylation ($\eta = 1$), all the available modification sites are equally accessible. Therefore, the methylation and demethylation rates are proportional to $(M - m)$ and $m$ respectively as given in Eq. (5). As a result, the adapted activity has a simple linear dependence on adapted methylation $\langle a \rangle^{\mathcal{A}} = (M - \langle m \rangle^{\mathcal{A}})/M$ as shown in Fig. 2b. An increase of ligand concentration from $[L]_1$ to $[L]_2$ triggers an immediate response (a drop in activity) followed by a slow adaptation process that leads the system to a different adapted activity level. The inaccurate adaptation for $\eta = 1$ is caused by the explicit dependence of $\langle a \rangle^{\mathcal{A}}$ on $\langle m \rangle^{\mathcal{A}}$, i.e., the solid line in Fig. 2b is tilted. It is easy to see that the dependence of $\langle a \rangle^{\mathcal{A}}$ on $\langle m \rangle^{\mathcal{A}}$ occurs for all $\eta \ne 0$.

Results from direct Monte-Carlo (MC) simulations of $\langle a \rangle$ subject to a series of step increases in concentration $[L]$ ($\times 10$ fold increase for each step) as shown in Fig. 2c for sequential and Fig. 2d for random methylation schemes support our analysis shown in Fig. 2a, b, respectively.

For sequential modification ($\eta = 0$), the adaptation error $\xi$ (see Methods section for its definition) is proportional to the probability of receptors in the extreme (boundary) methylation states $m = M$ & $0$ and we have:

$$\xi(\eta = 0) \approx c_1 \times \exp[-b^{-1}|\alpha|(M - m_0)] + \xi_0 , \quad (6)$$

where $c_1$ and $b$ are constants, $\xi_0$ is the error from the $m = 0$ state, and $\alpha$ ($<0$) is the free energy change for adding a methyl group to the receptor (see Eq. (10) in Methods section for the definition of $\alpha$). Equation (6) shows that $\xi$ decreases exponentially with $|\alpha|$ before saturating to $\xi_0$. For random modification ($\eta = 1$), the adaptation error has contributions from the whole range of methylation levels $0 \le m \le M$ and we have:

$$\xi(\eta = 1) \approx \frac{\ln(K_A/K_I)}{M|\alpha|} , \quad (7)$$

which only decreases with $|\alpha|$ algebraically (see Supplementary Note 1 for details of the derivation for Eqs. (6) and (7)).

The different dependence on $|\alpha|$ given in Eqs. (6) and (7) are verified by direct simulations (Supplementary Fig. 4a), which clearly show that sequential modification reduces adaptation error much more efficiently than random modification.

**The tradeoff between response gain and adaptation accuracy.** Besides the adaptation error $\xi$ or equivalently the adaptation accuracy $\xi^{-1}$, another important property of the system is its response gain $\Gamma$, which measures the sensitivity of the system in

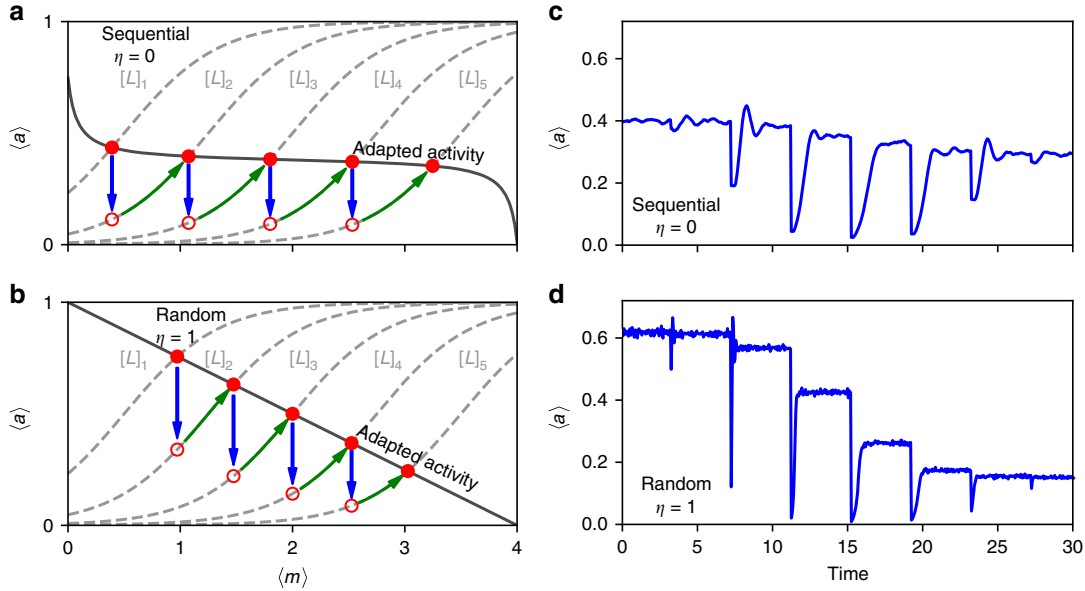

**Fig. 2 Response and adaptation to step changes in attractant concentration. a** Sequential ($\eta = 0$) and **b** random ($\eta = 1$) modification processes. The dashed lines show the dependence of the activity on $\langle m \rangle$ for different ligand concentrations $[L]_1 < [L]_2 < [L]_3 < [L]_4 < [L]_5$. The solid lines show the adapted activity $\langle a \rangle^{\mathcal{A}}$ as a function of the mean total methylation level $\langle m \rangle$. Upon a sudden increase in $[L]$, e.g., from $[L]_1$ to $[L]_2$, the system responds quickly by decreasing its activity (blue arrow) from the old adapted state (solid red circle) to the maximum response state (hollow red circle) without changing $\langle m \rangle$. This initial response is followed by the slow adaptation dynamics (green arrow) along the dashed line until the new adapted state is reached. Direct Monte-Carlo simulation results of the average activity $\langle a \rangle$ in response to a series of step increase in methyl aspartate concentration over 7 orders of magnitudes are shown for **c** Sequential ($\eta = 0$) and **d** Random ($\eta = 1$) modification processes. The step changes in stimulus is shown in Supplementary Fig. 8. The sequential modification process leads to a much higher adaptation accuracy than the random modification process.

response to a change in external signal (see Methods section for the definition of $\Gamma$).

As shown in Eqs. (6) and (7), adaptation accuracy can be increased by increasing $|\alpha|$, but what happens to the gain $\Gamma$? Interestingly, increasing $|\alpha|$ leads to a reduced gain independent of whether the modification dynamics is sequential or random (Supplementary Fig. 4b). The reason is that for a larger value of $|\alpha|$, individual receptors in the receptor cluster in the adapted state will have activities further away from the adapted mean value $\langle a \rangle \sim 1/2$—either closer to 0 or closer to 1—where the sensitivity (gain) is lower (see Supplementary Methods for details). It is worth noting that this dependence of $\Gamma$ on $|\alpha|$ is due to the discrete methylation level of individual receptor, which is only captured by the Ising model[19–21] but not in the simplified Monod–Wyman–Changeux (MWC) model[22–24].

The tradeoff or anti-correlation between response gain $\Gamma$ and adaptation accuracy $\xi^{-1}$ is a general property of the signaling pathway. Besides the extreme cases ($\eta = 0$ and $\eta = 1$) considered so far, this tradeoff between $\Gamma$ and $\xi^{-1}$ exists for all intermediate cases of methylation dynamics with $0 < \eta < 1$. As shown in Fig. 3a, the gain is almost unaffected when we change the value of $\eta$ while keeping the other parameters constant, but the corresponding adaptation accuracy $\xi^{-1}$ decreases with $\eta$. On the other hand, when we tune other parameters to maintain a high accuracy (e.g., by increasing $|\alpha|$), the corresponding gain goes down with $\eta$ as shown in Fig. 3b. Therefore, for a more random methylation scheme (a larger value of $\eta$), the tradeoff between $\xi^{-1}$ and $\Gamma$ means that one is enhanced at the expense of the other.

Accurate adaptation and high response gain represent two of the most desirable but opposing properties of biological signaling systems, i.e., to resist changes in the environment by adaptation and to respond to weak signals. This accuracy-gain tradeoff is related to the fluctuation–dissipation relationship established in equilibrium systems[25]. Next, we show how this tradeoff can be

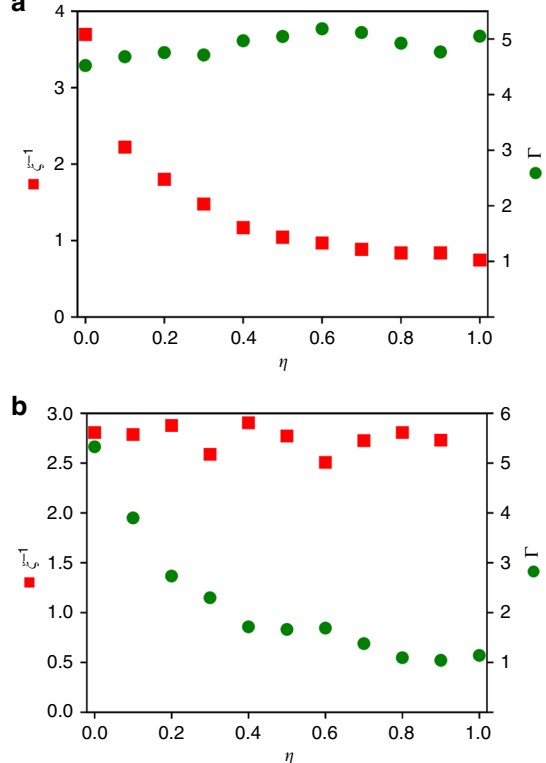

**Fig. 3 The tradeoff between the adaptation accuracy and response gain. a** As $\eta$ increases, the adaptation accuracy $\xi^{-1}$, red squares, decreases while the signaling gain $\Gamma$, green circles, remains roughly constant. **b** When parameters are tuned to keep the accuracy $\xi^{-1}$ roughly constant for different values of $\eta$, the corresponding gain $\Gamma$ decreases with $\eta$. The range of stimulus is set by $[L]_{min} = 1\,\mu M$ and $[L]_{max} = 100\,mM$.

attenuated by having multiple modification sites and sequential modification.

**Sequential modification attenuates the accuracy-gain tradeoff.** Why are there multiple modification sites in a regulatory protein or a receptor? How is the performance of the system enhanced by having multiple modification sites? Here, we investigate how having multiple modification sites affects the gain $\Gamma$ and accuracy $\xi^{-1}$ for different modification dynamics (sequential versus random).

For the sequential modification dynamics, the adaptation error comes from the receptor populations with the extreme (boundary) methylation levels $m = 0$ or $m = M$. As the probability $P_M$ of reaching the boundary state $m = M$ decreases exponentially with $M$, the adaptation error in the sequential modification model ($\eta = 0$) should decrease strongly (exponentially) with $M$ as given in Eq. (6). For the random modification dynamics, the adaptation error comes from all methylation levels and the reduction of adaptation error with increasing $M$ is much smaller $\sim 1/M$ as given in Eq. (7).

We studied the dependence of the performance of the system on $M$ systematically by computing $\Gamma$ and $\xi$ for a random set of parameters for $M = 1, 2, 3, 4$ in our models with $\eta = 0$ and $\eta = 1$. The results, as shown in Fig. 4, clearly demonstrate the general accuracy-gain tradeoff, i.e., the inverse dependence of $\Gamma$ and $\xi^{-1}$ in all cases studied. However, there are significant differences between the sequential and random modification cases. For sequential modification ($\eta = 0$), the tradeoff curve is lifted significantly as $M$ is increased as shown in Fig. 4a. In fact, the threshold lines (solid lines in Fig. 4), which are just fits to the highest performing points for each value of $M$, follow an approximate form:

$$\Gamma \ln(M\xi^{-1}) = C_0(M) , \qquad (8)$$

where $C_0(M)$ measures the overall performance of the system with sequential modification ($\eta = 0$). As shown in the inset of Fig. 4a, $C_0(M)$ increases significantly (linearly) with $M$. In contrast, as shown in Fig. 4b, the threshold lines in the random modification case follow a much more gradual curve:

$$\Gamma \xi^{-1} = C_1(M) , \qquad (9)$$

where the overall performance $C_1(M)$ for the random modification system ($\eta = 1$) has only a weak dependence on $M$ (see inset in Fig. 4b).

The significantly different dependence of the accuracy-gain tradeoff relationship on $M$ for $\eta = 0$ and $\eta = 1$ clearly shows that having multiple modification sites can ease the accuracy-gain tradeoff in general but the effect is significant only when the modification dynamics are sequential.

**Comparisons with existing experiments.** In this section, we discuss specific model results that can be directly compared with existing experiments. The *E. coli* chemoreceptor Tar has four methylation sites at residues 295, 302, 309, and 491, which are labeled by numbers 1–4, respectively. Protein methylation in eukaryotic cells is usually associate with lysine or arginine residues. However, glutamate is the most common residue for methylation in *E. coli*[26], and bacterial chemotaxis receptors in general are methylated in glutamate residues; or in glutamine residues that were posttranslationally deamidate to glutamates by CheB. Residues 1–3 are seven residues apart from each other, along the same $\alpha$ helix, whereas residue 4 is located on another helix[27] as illustrated in Fig. 5.

Experimental results[28] indicate that methylation of sites 1, 2, and 3 depends on each other in reverse order, i.e., site 3 is

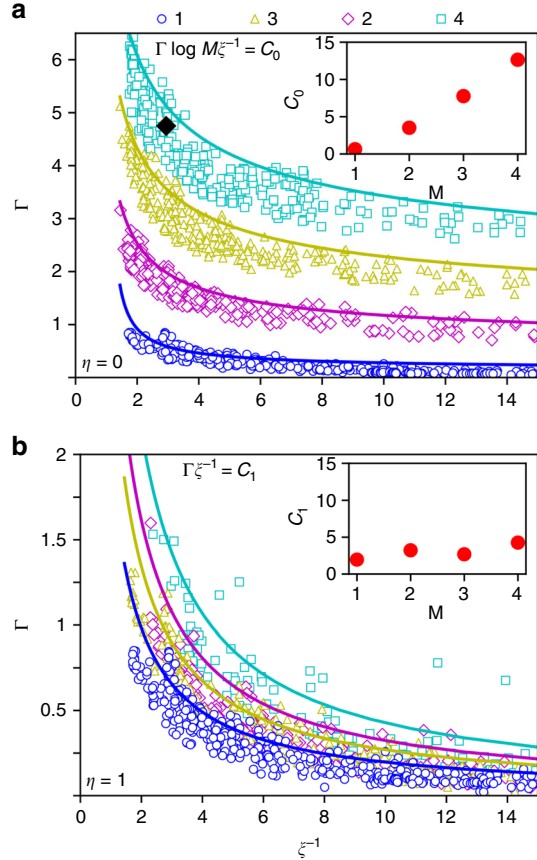

**Fig. 4 The gain-accuracy tradeoff.** The response gain $\Gamma$ versus the adaptation accuracy $\xi^{-1}$ for different parameters in **a** Sequential ($\eta = 0$) and **b** random ($\eta = 1$) modification models. Each symbol corresponds to a different choice of parameters ($m_0$, $\alpha$, $M$) with its color corresponding to the value of $M = 1$ (blue circles), 2 (red diamonds), 3 (yellow triangles), 4 (light blue squares). For each value of $M$, the best performing points were fitted with the expressions given in Eqs. (8) and (9) as shown by the solid lines. The fitted values of the overall performance parameters $C_0$ and $C_1$ for different values of $M$ were plotted in the insets. The black diamond is obtained from experimental results for Tar receptor in response to methyl aspartate[32,33].

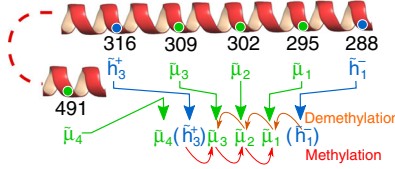

**Fig. 5 Illustration and notation for the Tar receptor.** The methylation state of site $i$ (=1, 2, 3, 4), green circles, is described by a binary variable $\tilde{\mu}_i$ (1—methylated; 0—unmethylated). The sequential methylation and demethylation processes among sites 1-2-3 are shown by the red and orange arrows. The two initiator sites (316 and 288), blue circles, are described by two binary numbers $\tilde{h}_3^+$ and $\tilde{h}_1^-$, which control the initialization of the sequential methylation and demethylation processes: $\tilde{h}_3^+ = 1$ promotes methylation of site 3; $\tilde{h}_1^- = 0$ promotes demethylation of site 1.

methylated first, followed by site 2 and then site 1, and that residues 316 and 498 affect the methylation of site 3 and 4, respectively. Structural models[29] of the receptor modification are consistent with the methylation rate depending on a residue seven residues away in the C-terminal direction. The initiator residues,

labeled with the letter $\tilde{h}$ in Fig. 5, help initialize the sequential methylation or demethylation processes, but are not themselves methylation sites. The initiators of the reverse sequential methylation in bacterial receptors encompassing sites 1, 2, and 3 are represented by $\mu_0 \equiv \tilde{h}_3^+$ with $j = 1$ in Eq. (2) and $\mu_4 \equiv \tilde{h}_1^-$ with $j = 3$ in Eq. (3).

The receptor methylation state is described by six binary numbers, $\tilde{\mu}_4(\tilde{h}_3^+)\tilde{\mu}_3\tilde{\mu}_2\tilde{\mu}_1(\tilde{h}_1^-)$. In our notation, a methylation site is modifiable when it is labeled by $x$ and a specific value (0 or 1) is assigned when it is fixed by mutation. The two initiator residues are given by $\tilde{h}_3^+$ and $\tilde{h}_1^-$—when $\tilde{h}_3^+ = 1$, methylation at site 3 ($\tilde{\mu}_3$) becomes enhanced; when $\tilde{h}_1^- = 0$, demethylation of site 1 ($\tilde{\mu}_1$) becomes enhanced. Otherwise when $\tilde{h}_3^+ = 0$ or $\tilde{h}_1^- = 1$, the initial methylation of site 3 or the initial demethylation of site 1 are controlled by the slow random methylation or demethylation processes.

We note that though the existence of the $\tilde{h}_3^+$ site is supported by experiments[28–30], the initiator site $\tilde{h}_1^-$ for demethylation is introduced here hypothetically according to the close relationship between the two enzymes CheR and CheB as suggested in a study by Djordjevic et al.[31], which stated that "structural similarity between the two companion receptor modification enzymes, CheB and CheR, suggests an evolutionary and/or functional relationship" and "the proposed receptor interaction clefts occur on different faces of the $\beta$-sheet in CheB and CheR. Topological differences in the structures of CheR and CheB may be reflective of their functionally antagonistic interactions with the receptors".

We first study the adaptation accuracy and response gain from our model and compare them with available experiments. For wild-type (wt) cells with both CheR and CheB, though there is no direct measurements of the methylation dynamics, there have been detailed experimental studies of the in vivo kinase activity dynamics in response to a wide range of stimuli[32–34], which can be compared with our model to determine the response gain $\Gamma$ and adaptation accuracy $\xi^{-1}$.

In ref. [32] the relative sensitivity, $S_r$, is defined as the fractional change in the FRET signal divided by fractional change in stimulus $S_r = \frac{\Delta\text{Fret}/\text{Fret}}{\Delta[L]/L} \sim g$, where the FRET signal is proportional to the kinase activity and $g$ is the integrand of Eq. (17). From the experimental data on $S_r$ (first peak in Fig. 3 of ref. [32]) and our model, we can estimate the value for the gain $\Gamma$ for Tar: $4.5 \lesssim \Gamma \lesssim 5$.

From the measured adapted activity for different background ligand concentrations as plotted in Fig. 1B in the paper by Neumann et al.[33], we obtained the value of adaptation accuracy for Tar in response to methyl aspartate with a maximum concentration $[L]_{max} = 5\text{--}10$ mM to be roughly in the range: $2.3 \leq \xi^{-1} \leq 3.5$.

These estimated values of gain and adaptation accuracy, shown as the black diamond in Fig. 4a, suggest that methylation dynamics should be mostly sequential. Quantitatively, from our model and by using the values of $\xi$ and $\Gamma$, we can determine the range of the effective randomness parameter for Tar: $0.05 \leq \eta \leq 0.13$ (see Supplementary Methods and Supplementary Fig. 2 for more details on comparison between simulation results and experiments).

Next, we study the methylation profiles in CheB$^-$ mutants by using our model and compare them with existing experiments. Different modification dynamics, random or sequential, lead to qualitatively distinct mean methylation profiles at a given time. Denote $p_j(t)$ the probability of site $j$ being methylated at time $t$. For a purely random modification dynamics, all $p_j(t)$ should be the same. However, for methylation dynamics that are dominated

**Table 1 Experimentally measured methylation rates.**

| Mutant | 1 | 2 | 3 | 4 | Simulation |
|---|---|---|---|---|---|
| EEEE | 0.03 | 0.28 | 0.64 | 0.05 | 0(1)xxx(0) |
| EEE(N)E | 0.02 | 0.19 | 0.38 | 0.01 | |
| EEQE | 0.15 | 0.74 | — | 0.10 | 0(1)1xx(0) |
| EEDE | 0.044 | 0.017 | — | 0.08 | 0(1)0xx(0) |

Normalized methylation rates for different CheB$^-$ mutants reproduced from refs. [28,30]. The four letters (D for aspartate, E for glutamate, N for asparagine, and Q for glutamine) in the first column are the residues, respectively, in the methylation sites 1, 2, 3, and 4. The four middle columns are the methylation rates of each site in arbitrary units. Simulations mimicking each mutant were performed with their configurations shown in the last column, with x representing modifiable sites. We fixed $\tilde{\mu}_4 = 0$ due to the low methylation rate of site 4.

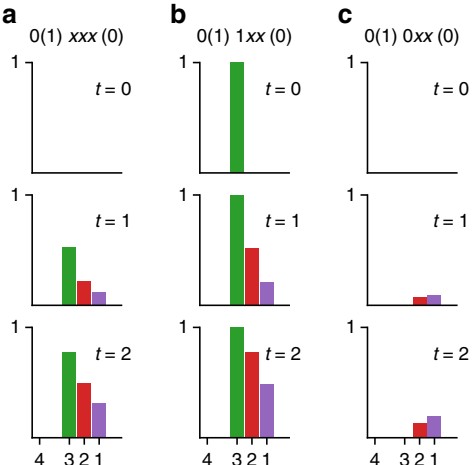

**Fig. 6 Methylation dynamics of the CheB$^-$ strains from the model.** The predicted methylation dynamics in response to a saturating level of attractant for three different CheB$^-$ mutants listed in Table 1: **a** 0(1)xxx(0); **b** 0(1)1xx(0); **c** 0(1)0xx(0). The probabilities of being methylated at each methylation site (horizontal axis) are shown at different time $t$. The parameters used here are $\eta = 0.1$, $k^+ = 0.7$ s$^{-1}$, and $k^- = 0$.

by sequential modification, the methylation pattern among different sites follows certain distinctive patterns.

The methylation dynamics among different sites in the demethylase CheB$^-$ mutants were studied by the Koshland lab more than 20 years ago[28,30], by mutating the residues $\tilde{\mu}_4$, $\tilde{h}_3^+$, $\tilde{\mu}_3$, $\tilde{\mu}_2$ in Fig. 5, and the initiator $\tilde{h}_4^+$. After being methylated with tritiated SAM, the receptors were cleaved and the extent of methylation of each site was determined by high-performance liquid chromatography. The methylation rates were calculated in arbitrary units, reproduced here in Table 1. As the absolute values are not available, we can only analyze the relative methylation ratios of the different sites and mutants.

It is useful to compare simulations with $k^- = 0$ with the CheB$^-$ mutants to isolate the effects of sequential methylation. Specifically, these mutants are, besides the wild-type receptor (EEQE), the mutant receptors EEDE and EEEE. In these strains, site 3 (E309) can be methylated when occupied by Glutamate (E) residues, but behave as permanently methylated or demethylated when occupied, respectively, by aspartate (D) or glutamine (Q). In addition, substitution of $\tilde{h}_3^+$ by asparagine (N) in mutant E(N)EEE is also informative as it partially impairs the methylation of site 3, which would correspond to a partial methylation state in our model.

We studied the methylation dynamics of these $CheB^-$ mutants by using a sequential-dominant model with a small value of $\eta$ (= 0.1). As shown in Fig. 6a, the sequential methylation of a completely demethylated receptor $(0(1)xxx(0))$ begins by methylating site 3, afterwards proceeds to site 2, and to site 1. This order, i.e., $p_3(t) > p_2(t) > p_1(t)$, persists throughout the methylation process consistent with experiments results. We also studied the methylation dynamics when the starting site $\tilde{\mu}_3$ is fixed to be $\tilde{\mu}_3 = 0$ and $\tilde{\mu}_3 = 1$ to mimic the *EEDE* and *EEQE* receptors, respectively. As shown in Fig. 6b, when we fix $\tilde{\mu}_3 = 1$, the order of methylation for site 2 and site 1 still persists, i.e., $p_2(t) > p_1(t)$, which is again consistent with experiments.

Finally, the most informative and also most stringent test of our theory comes from the mutant receptor *EEDE*. Besides a much slower methylation rate, an inverted behavior $p_2 < p_1$ was observed experimentally in *EEDE* (Table 1). Remarkably, these behaviors in particular the inversion also appear in our model as shown in Fig. 6c. The reason for this inversion is that sequential modification is broken when site 3, the starting site in the sequence, cannot be methylated. As a result, the downstream sites (site 2 and site 1) have to be methylated (at least initially) by the random methylation process, which has no a priori preference between site 1 and site 2. Once site 2 becomes methylated, it will enhance the methylation rate at site 1 due to the sequential methylation process but not the other way around. Thus the sequentiality between site 2 and site 1 leads to the observed inversion. Consistent with this argument and with the role played by the initiator, the partial methylation of $\tilde{h}_3^+$ in *EEE(N)E* reduces the ratio $p_3/p_2$, when compared to *EEEE*. Quantitatively, the $CheB^-$ data lead to an estimate for a lower bound for the random methylation parameter $\eta \geq 0.047$ (see Supplementary Discussion for more details), which is consistent with the estimated range of $\eta$ for *Tar* from the wt data above.

Overall, our model results, together with existing experimental data, suggest that the methylation process for sites 3, 2, and 1 are mostly sequential and are affected by the initiator $\tilde{h}_3^+$, but there is a small but finite random component.

**Testable predictions for future experiments**. Our model can be used to predict the methylation level profile for different mutants, which can be tested by future experiments. As the reference, the methylation levels of the wild-type cell $(0(1)xxx(0)$ receptors in the presence of *CheR* and *CheB*) decrease monotonically from site 3 to 1 as shown in Fig. 7a. We first study the mutant with $\tilde{h}_3^+ = 0$, which inhibits sequential methylation of site 3. As shown in Fig. 7b, $\tilde{h}_3^+ = 0$ brings down the methylation of site 3, leading to site 2 being more methylated than sites 1 and 3. To explore the inhibition of sequential demethylation of site 1, we next study the

mutant with $\tilde{h}_1^- = 1$. As shown in Fig. 7c, site 2 is less methylated than sites 1 and 3 in the steady state. Finally, we study the mutant with $\tilde{h}_1^- = 1$ and $\tilde{h}_3^+ = 0$ in which both methylation of site 3 and demethylation of site 1 are inhibited. As shown in Fig. 7d, the steady state methylation profile monotonically increases from 3 to 1, which is exactly the inverse of the wt profile (Fig. 7a).

We can also predict the effects of mutating the key methylation sites ($\tilde{\mu}_1$ and $\tilde{\mu}_3$) on adaptation dynamics. We first studied effects of mutating site 1 or site 3 to be permanently unmethylated by fixing either $\mu_3 = 0$ or $\mu_1 = 0$ in our model. We found that adaptation still works in $\mu_3 = 0$ mutant $[0(1)xx0(0)]$ but is severely impaired in the $\mu_1 = 0$ mutant $[0(1)0xx(0)]$ as shown in the Supplementary Fig. 7a. We next studied effects of mutating site 1 or site 3 to be permanently methylated by fixing either $\mu_3 = 1$ or $\mu_1 = 1$ in our model. We found that response to decrease in attractant concentration remains intact in the $\tilde{\mu}_3 = 1$ mutant $[1(1)1xx(0)]$, but is severely impaired in the $\tilde{\mu}_1 = 1$ mutant $[1(1)xx1(0)]$ as shown in Supplementary Fig. 7b. These predictions can be tested by measuring the kinase activity dynamics in vivo in these mutants by using FRET[32].

## Discussion

Multisite regulatory proteins are ubiquitous in biology, yet their functions are not well understood. Here, we studied effects of ordering among the multiple modification sites and possible benefits of having multiple sites in the context of bacterial chemotaxis. We discuss the two main findings below.

First, we found that sequential modification is crucial for perfect adaptation. Previous study[14,35] showed that perfect adaptation can be achieved by an integral control mechanism where dynamics of the controller (receptor methylation level) only depend on receptor activity. Here, we showed that sequential modification is another important ingredient for the integral control mechanism as it guarantees the methylation/demethylation rates to be independent of the receptor methylation level, Eq. (4). As a direct consequence of sequential modification, the adapted activity is independent of the receptor methylation level (or the stimulus strength), i.e., perfect adaptation.

We note that there may be other possible scenario for the methylation/demethylation rates to be independent of the available modification sites. For bacterial chemoreceptors, the binding and unbinding of CheR to the receptor are faster than its catalytic rate and the dissociation constant $K_D$ is relatively small[36]. If the enzyme binds to all available active sites randomly with equal probability, the number of available sites effectively changes the substrate concentration. Given that the substrate concentration is much higher than the Michaelis–Menten constant $K_M \approx K_D$, the methylation reaction rate, which is limited by the slow catalytic reaction, would be independent of the substrate concentration and thus independent of the number of available methylation (demethylation) sites. However, the binding rate ($k_{on}$) of the enzyme in this scenario would depend on the substrate concentration and the available modification sites, which seems to be inconsistent with the recent in vitro measurements of the $k_{on}$ rates for CheR binding to Tar(*EEEE*) and Tar(*QQQQ*) receptors[36] (see Supplementary Note 1 for more details). Furthermore, the random methylation pattern predicted by this scenario is inconsistent with the observed sequentiality among the different methylation sites in in vivo experiments[28,30].

Second, we found that there is a tradeoff between response gain and adaptation accuracy. We showed that this tradeoff can be improved significantly by having more modification sites but only with the sequential modification process. Taken together, our study suggests a general two-pronged strategy to enhance chemotaxis performance by having multiple modification sites to

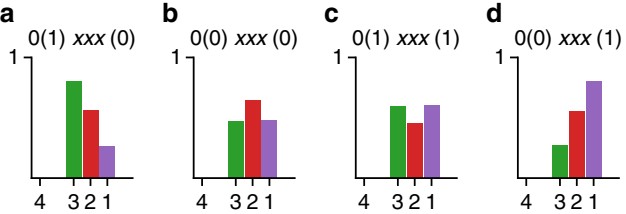

**Fig. 7 The predicted methylation distributions.** The steady state methylation level distributions of the Tar receptor for four different strains with different initiator sites ($\tilde{h}_3^+$ and $\tilde{h}_1^-$) predicted from our model: **a** Strain $0(1)xxx(0)$ with $\tilde{h}_3^+ = 1$ and $\tilde{h}_1^- = 0$; **b** strain $0(0)xxx(0)$ with $\tilde{h}_3^+ = 0$ and $\tilde{h}_1^- = 0$; **c** strain $0(0)xxx(1)$ with $\tilde{h}_3^+ = 0$ and $\tilde{h}_1^- = 1$; **d** strain $0(1)xxx(1)$ with $\tilde{h}_3^+ = 1$ and $\tilde{h}_1^- = 1$. Parameters used here are $\eta = 0.1$, $[L] = 100\ \mu M$, $k^+ = 0.7\ s^{-1}$, and $k^- = 1.4\ s^{-1}$.

extend the dynamic range of high gain, and a sequential modification process to maintain adaptation accuracy. Direct comparison with existing experiments confirms our theory and reveals that the methylation process for methylation sites 3, 2, and 1 of Tar is mostly sequential with a small but finite random component $0.05 \leq \eta \leq 0.13$. The confirmed importance of sequential receptor methylation begs the question of the underlying molecular mechanism responsible for maintaining specific ordering in multisite modification. Previous mutant studies showed that methylation of a given site is affected by a residue seven amino acids to the C terminus[28,30], which is exactly how sites 1, 2, and 3 are arranged (Fig. 5). Also, methylation of site 3 is affected by a residue 7 amino acids to the C terminus, even though that residue itself is not a methylation site[28]. Indirect evidence of sequential demethylation by cheB can also be found in refs. [37,38] (see Supplementary Discussion for details).

The existing experiments mentioned above suggest a chain reaction scheme for the sequential methylation process. However, it is not clear whether the preceding site in the sequence increases the binding affinity of CheR to the receptor or the catalytic rate or both. It is also not clear whether and how different receptors in the closely packed receptor cluster compete for the limited CheR molecules in the cluster. We believe that a detailed biochemical model that incorporates key steps such as binding/unbinding and catalytic reactions in the methylation/demethylation processes together with quantitative in vitro measurements of the methylation/demethylation rates for wt and mutant receptors are needed to address these questions. The same strategy should also be used to study the much less known demethylation process. In addition to searching for possible molecular mechanisms for ordered modification, another interesting question is what are the thermodynamic costs of implementing such ordered modification mechanisms for accurate control[24]. Finally, it is worth pointing out that even though the detailed molecular mechanism of the methylation and demethylation reactions still remains open, our conclusions regarding the general properties of the system such as response gain, adaptation accuracy, and their tradeoff and their dependence on the level of sequentiality ($\eta$) of the underlying multisite modification process should hold true.

Our work serves as a successful case study of multisite protein modification by using a modeling approach in combination with knowledge of the underlying biochemical pathway and quantitative data. This combined approach provides a powerful general framework that can be applied to other signaling systems to understand the mechanisms of multisite signaling proteins and their biological functions.

## Methods

**The standard model for bacterial chemotaxis.** We briefly describe a previously developed general mathematical framework—the standard model for studying bacterial chemotaxis signaling pathway dynamics (see ref. [17] for a recent review).

In the standard model for bacterial chemotaxis, each receptor has two key state variables—its kinase activity ($a$) and its methylation state ($\mu$). For kinase activity, a receptor can be either active ($a = 1$) or inactive ($a = 0$). For methylation state, as each receptor has $M (\geq 1)$ modification (methylation) sites, there are a total of $2^M$ possible modification states characterized by a $M$ − dimensional binary vector $\mu = (\mu_1, \mu_2, \ldots, \mu_M)$, where the binary number $\mu_j = 0, 1$ respectively represents the unmethylated and methylated state of site $j (=1, 2, \ldots, M)$. The total modification level of a receptor is given by: $m \equiv \| \mu \| = \sum_{j=1}^{M} \mu_j$.

The receptor kinase activity dynamics is fast relative to its methylation dynamics. Here, we use the standard two-state model to describe the receptor kinase activity dynamics, where the active and inactive states are separated by a free energy difference $\Delta f$. When the fast ligand–receptor binding/unbinding process is averaged out, $\Delta f(m, [L])$ depends on the receptor's total modification level $m$ and the ligand concentration $[L]$. From previous studies on bacterial chemotaxis[18,35], the free energy $\Delta f$ can be written as:

$$\Delta f(m, [L]) = -\ln \left( \frac{1 + [L]/K_I}{1 + [L]/K_A} \right) - \alpha(m - m_0) , \quad (10)$$

where $K_I$ and $K_A$ are the dissociation constants of the ligand binding to the inactive and active conformations of the receptor, $\alpha (<0)$ is the free energy change due to adding (or removing) one methylation group to the receptor, and $m_0$ determines the average modification level in the absence of any stimulus ($[L] = 0$).

Another important phenomenon in bacterial chemotaxis is that bacterial chemoreceptors form polar clusters[39–41]. The receptors and their kinase activities are coupled with each other in the cluster. Following previous work[19,42], we model the receptor cluster by using an Ising-type model with nearest neighbor interaction with strength $C$.

Dynamic Monte-Carlo simulations of the Ising-type model (see Supplementary Methods for details of the Monte-Carlo simulations) is used to obtain the distribution of receptors, $P_{a\mu}$, in a given state ($a\mu$), which describes the statistical properties of the receptor cluster. From the full distribution function $P_{a\mu}$, distribution of the microscopic methylation state $\mu$ can be obtained by summing over the fast variable $a$, $P_\mu = \sum_{a=0}^{1} P_{a\mu}$, and the probability of the total modification level $m$ is given as:

$$P_m = \sum_{\|\mu\|=m} P_\mu = \sum_{\|\mu\|=m} \sum_{a=0}^{1} P_{a\mu} . \quad (11)$$

From these distribution functions, average properties of the receptor cluster can be obtained. For example, the average methylation level is

$$\langle m \rangle = \sum_m m P_m . \quad (12)$$

According to Eq. (10), kinase activity of a receptor $\langle a \rangle_m$ only depends on its total methylation level $m$, which can be expressed as:

$$\langle a \rangle_m = \frac{1}{P_m} \sum_{\|\mu\|=m} P_{1\mu} , \quad (13)$$

and the average activity for all receptors is:

$$\langle a \rangle = \sum_m \langle a \rangle_m P_m . \quad (14)$$

These distribution functions and average receptor properties are used here to understand the response gain and adaptation accuracy in bacterial chemotaxis quantitatively. In particular, we focus on investigating how different modification schemes (random or sequential) affect the adaptation accuracy and response gain in this paper.

**Characterizing the performance of the chemotaxis signaling pathway.** The performance of the chemotaxis signaling pathway can be characterized by two key system-level properties: the integrated response gain (amplification) $\Gamma$ and adaptation accuracy $\xi^{-1}$, which we define in the following.

At a given background ligand concentration $[L]$, the adapted methylation levels of all receptors in the system (receptor cluster) are represented by $\mathbf{m}^{\mathcal{A}}([L])$ (vector $\mathbf{m} = (m_1, m_2, \ldots, m_N)$ contains the methylation levels of all the receptors in the receptor cluster, $m_i$ is the methylation level of receptor-$i$ ($1 \leq i \leq N$) and $N$ is the number of receptors in the cluster) and the average adapted activity of the system is given by: $\langle a \rangle^{\mathcal{A}}([L]) \equiv \langle a \rangle (\mathbf{m}^{\mathcal{A}}([L]), [L])$. Upon a sudden change of ligand concentration from $[L]$ to $[L] + \delta[L]$, the system first responds by a change of activity $\delta \langle a \rangle$, which can be written as:

$$\delta \langle a \rangle \approx \langle a \rangle (\mathbf{m}^{\mathcal{A}}([L]), [L] + \delta[L]) - \langle a \rangle (\mathbf{m}^{\mathcal{A}}([L]), [L]) \approx \frac{\partial \langle a \rangle}{\partial [L]} \bigg|_{\mathbf{m}^{\mathcal{A}}([L])} \times \delta[L] \quad (15)$$

as the methylation levels remain approximately unchanged at their pre-stimulus levels $\mathbf{m}^{\mathcal{A}}([L])$ due to their slow dynamics. Following Sourjik and Berg[32], we define the response gain at the background $[L]$ as the ratio of fractional changes in activity and ligand concentration:

$$g([L]) \equiv -\frac{\delta \langle a \rangle / \langle a \rangle^{\mathcal{A}}([L])}{\delta[L]/[L]} \approx -\frac{[L]}{\langle a \rangle^{\mathcal{A}}([L])} \frac{\partial \langle a \rangle}{\partial [L]} \bigg|_{\mathbf{m}^{\mathcal{A}}([L])} , \quad (16)$$

where the negative sign is due to the fact that increase of attractant concentration leads to decrease of receptor activity in bacterial chemotaxis.

To describe the system's ability to amplify the input stimulus over a broad range of background stimulus concentration, we define the overall gain $\Gamma$ as the integral of $g([L])$ over the range of $[L]$ (in log-scale to capture the broad range of ligand concentration):

$$\Gamma = \int_{[L]_{min}}^{[L]_{max}} g([L]) \, d\log_{10}[L] . \quad (17)$$

In our simulations, we compute the response gains by using a small but finite fractional change $\delta[L]/[L] = \pm 0.1$ at a series of background ligand concentrations $[L] = [L]_i$ ($i = 1, 2, \ldots$) that are equally spaced in log-scale to cover the whole response range. The overall gain $\Gamma$ is obtained by summing the gains at different background levels (see details of computing $\Gamma$ in Supplementary Methods).

As the adapted activity only depends on the ligand concentration $[L]$, we define the adaptation error as the relative change of adapted activity for an infinitesimal

fractional change of $[L]$, i.e.,

$$\epsilon([L]) \equiv -\frac{[L]}{\langle a \rangle^{\mathcal{A}}([L])}\frac{d\langle a \rangle^{\mathcal{A}}([L])}{d[L]} \ . \tag{18}$$

To characterize the adaptation accuracy over the wide range of backgrounds, we define an overall adaptation error $\xi$ by integrating $\epsilon$ over the stimulus concentration in log-scale (natural base is used here for convenience):

$$\xi = \int_{[L]_{\min}}^{[L]_{\max}} \epsilon([L]) d\ln[L] = \ln\frac{a^{\mathcal{A}}([L]_{\min})}{a^{\mathcal{A}}([L]_{\max})} \ . \tag{19}$$

The overall adaptation accuracy is defined as the inverse of the adaptation error, $\xi^{-1}$. Details of the Monte-Carlo simulations and of computing $\Gamma$ and $\xi$ are discussed in Supplementary Methods.

**Reporting summary**. Further information on research design is available in the Nature Research Reporting Summary linked to this article.

## Data availability

All data used to support the findings of this work are available upon request.

## Code availability

The code used to perform the simulations is available at https://github.com/bernardomello/chemotaxis.

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

## Acknowledgements

We thank Drs. Xiaojing Yang and Chao Tang for discussions on Sic1 signaling in yeast cell cycle control. This work is partially supported by NIH grants (R01GM081747 and R35GM131734 to Y.T.).

## Author contributions

Y.T. and B.A.M. initiated the project. B.A.M. and A.B.B. developed the code and ran the simulations. Y.T. and B.A.M. did the analysis. Y.T. and B.A.M. wrote the paper.

## Competing interests

The authors declare no competing interests.
