## [Peer Review File · Nature Communications]

Reviewers' comments:

Reviewer #1 (Remarks to the Author):

This manuscript reports a model for sequential methylation of bacterial chemoreceptors. It has been appreciated for some time that the high precision of adaptation observed for the chemotaxis system of *E. coli* implies that adaptive methylation/demethylation rates should depend only on receptor activity. While some studies have addressed deviations from precise adaptation and incorporated the limits of adaptation set by the number of modification sites, there has been no clear picture of what mechanism could be responsible for methylation/demethylation rates being (largely) independent of existing methylation levels. The authors make a compelling proposal that sequential methylation could be this underlying mechanism. Importantly, they support their suggestion by careful consideration of previous experimental observations in Refs. 30 and 31: (1) modification rates at a given site depend on the identity of a residue two helix turns away that would be altered during the course of methylation/demethylation, and (2) mutations to these "regulatory" residues led to differential methylation/demethylation rates that are consistent with the sequential modification hypothesis. The authors also explore the implications of sequential modification, and show that the model is consistent with observations of both precision of adaptation and of response gain. Finally, the authors make additional testable predictions. This work addresses a long-standing mystery in the field of bacterial chemotaxis and will therefore likely be of significant interest to the readership of *Nature Communications*. However, there are a number of issues that must first be addressed.

Major issue:

1. Necessity for a sequential model. While the authors' sequential model is very appealing, it is not the only possible mechanism that could underly precise adaptation. Any mechanism for which methylation/demethylation rates depend only on receptor activity leads to precise adaptation. E.g. any mechanism in which methylation/demethylation rates are limited by local CheR/CheB activity would behave similarly. I.e. if CheR and CheB bind quickly to available sites, but their catalytic activity is rate limiting, then this would also lead to overall modification rates independent of current methylation levels. As written, the manuscript repeatedly states that sequential modification is the only possible way of achieving precise adaptation, but this is not correct. Given the centrality of this point to the paper, it is essential that the authors acknowledge that there are other ways that precise adaptation might be achieved. Along these lines, the authors need to revise the presentation of their model, which actually only represents a narrow class of possible models.

Minor issue:

2. In S4, third line "Fig. 2b,c close to each other..." I believe should be "Fig. 2a,b..." and "stepper" -> "steeper".

Reviewer #2 (Remarks to the Author):

Sequential modification of bacterial chemoreceptors is key for achieving accurate adaptation and high gain simultaneously

Mello, Beserra and Tu

Nature Communications paper # NCOMMS-19-27808

General comments

Let me begin by apologising for the delay in completing this review, which was due to some unexpected personal issues.

This is an interesting paper which addresses two issues of broad interest for understanding cellular information processing. First, what is the functional benefit of having multiple sites of protein post-translational modification (PTM)? Second, given that there are multiple sites on a protein, is there a functional advantage in enzymatic modification and demodification being sequential? The answers offered here are that multisite PTM of receptor proteins enables chemotaxis in *E. coli* to be both sensitive and adaptive and that sequentiality of multisite PTM supports this and may be necessary for it (below). Sequentiality has frequently been assumed in mathematical models, purely for convenience and with little experimental justification, so it is of considerable interest that this paper suggests that it might actually be favoured biologically. Overall, this is an important study which builds on a substantial body of previous work on bacterial chemotaxis by the senior author. However, there are issues that need clarifying, primarily to do with explaining the underlying mathematical model. In this respect, I was unable to work out exactly what the authors were doing and this problem ought to be fixed. I hope the comments below are of some help in doing so.

I believe the review process is improved by transparency and therefore do not wish to remain anonymous.

Jeremy Gunawardena

Specific comments

(1) Model details. Previous studies have used the sequential model, whose methylation states are shown in Fig.1a. The present paper is noteworthy in considering also the so-called “random” model in Fig.1b. However, I could not find a mathematical treatment of the random model in the main text or the SI. Citation 14 is given as a reference but this review paper appears to only treat the sequential model. The random model has a lot of moving parts: an individual receptor has, in addition to the methylation states shown in Fig.1b, two additional binary state variables, given by whether or not ligand is bound and whether or not the receptor is active and the receptor cluster is a population of interacting receptors. These additional degrees of freedom are averaged over in the present paper through a timescale separation, in which binding, activation and receptor interactions are treated as fast compared to methylation dynamics. This may be reasonable but it means that there is a “black box” whose inner workings are passed off to reference 14 which, as noted above, does not, in fact, work through the details of the random model. I found the lack of mathematical details problematic for several reasons.

1. **It is not clear how the key equations (Eq.4) are derived.** Reference is made to the SI but §S1 jumps from the master equation for methylation states only (Eq.S1) to equations for the total methylation level, m , (Eq.S2) without explaining how to get from the former to the latter and without introducing the activities which appear in Eq.4. Moreover, the equations are only stated for $\eta = 0$ (Eq.S2a) and $\eta = 1$ (Eq.S2b), which is strange when the paper discusses the full range $0 \leq \eta \leq 1$. I presume the authors have worked these matters out in full, so I do not understand why they have not provided the details. I appreciate that there are notational headaches to overcome when describing random models of this kind but that is what the SI is for, to avoid frightening readers who are “*less mathematically inclined*”. For similar kinds of models, we have found that using set notation makes it much easier to express the mathematical details; see, for instance, PMID 27368104 for binding models and PMID

29945451 for describing PTM patterns.

2. **As a consequence of the previous point, the underlying assumptions are not specified clearly and this makes the conclusions of the paper less compelling.** For instance, it seems to follow from Eqs.2 and 3 that there are only two free parameters for the methylation dynamics, k^+ and k^- , but the forward modification and backward demodification rates are scaled in terms of the quantity $\langle a \rangle_m$. I can see why this might help in deriving Eq.4 but for an enzyme to sense something that depends on the state of the entire receptor cluster seems like a strong assumption. Is the enzyme sampling the cluster in some way? Moreover, is it reasonable to claim that a sequential model is necessary for the conclusions that are drawn when the random model to which it is being compared is so special? With more subtle assumptions, perhaps a random model might also be capable of approximately perfect adaptation? At least, how can this be ruled out?

This is an important point because the necessity for sequentiality is one of the main contributions of the paper. It is a surprising result because the feedbacks are all the same, so why should the presence of non-sequential methylation patterns compromise perfect adaptation? I would like to have a clear reason for this but was unable to disentangle one from the text. I strongly encourage the authors to provide it.

3. **The notation is confusing.** The “state” of the random model sometimes refers to the pattern of methylations on the receptor, $\vec{m} = (m_1, \dots, m_4)$, and sometimes to the total methylation level, $m = \sum_i m_i$, but these symbols are not used consistently. For instance on page 4 (para 4, lines 1 and 2), “For the general case, \dots state- m is given by \dots ”, seems to refer to the random (general) model but its state is described by m rather than by \vec{m} . I suggest carefully distinguishing these two cases or, preferably, not using the same letter for both \vec{m} and m . The problem becomes acute with Eq.4b. What does a_m mean in this formula? It occurs inside the average, so it presumably refers to an individual receptor. But how does a receptor activity, a , depend on a total methylation level m ? If mathematical details had been given, it might have been possible to work this out but I am afraid I am stumped.

(2) **Gain and error.** The meaning of, and the relationship between, these two properties is hard to grasp. It would help to explain perfect adaptation before defining the adaptation error (Eq.6). If that had been done, it would be easy to justify the error as the difference between the adapted activities at the ends of the range, as on the RHS of Eq.6. It would not be necessary to define it in terms of a logarithmic derivative which is then integrated, as on the LHS of Eq.6. This derivation seems unnecessarily complicated and the integral formula is particularly strange because it looks very much like the formula for the gain given in Eq.5. The relationship would be even clearer if, in Eq.5, the lower and upper limits were written correctly, and, in Eq.6, $\langle a \rangle^{\text{adapt}}[L]$ was written in place of $\langle a \rangle^{\text{adapt}}$ and \log_{10} in place of \log . The difference between these two formulas then lies in the terms,

$$\left. \frac{\delta \langle a \rangle}{\delta [L]} \right|_{(m^{\text{adapt}}[L])} \quad \text{and} \quad \frac{d \langle a \rangle^{\text{adapt}}}{d [L]},$$

in Eq.5 and Eq.6, respectively. The latter is straightforward: the average activity after adaptation is a function of the ligand concentration and the derivative is being taken. As for the former, it is stated that $\delta \langle a \rangle$ is the “immediate (fast) response \dots resulting from a small change $\delta [L]$ in the stimulus”. There are several things I do not understand about this.

1. The meaning of the former quantity now depends on the timescale separation that leads to the “black box” referred to in point (1), whose details have not been given. It appears to be a transient, time-dependent quantity (“fast”) in contrast to the latter quantity, which is a steady-state value. If so, at what time is the value being determined? No time is given in the formula.

2. Why is δ used here, in place of d ? The implication is that it is a finite difference rather than an infinitesimal derivative. But no higher-order terms are present in the formula, so why is it necessary to make this distinction?
3. What does the limitation to $m^{\text{adapt}}[L]$ mean? Normally, such a limitation specifies the value of the independent variable, which in this case is $[L]$. But a total methylation level cannot be a ligand concentration. Again, I am stumped.

(3) Experiments and predictions. The comparisons to experimental results are appealing and substantially enhance the paper. However, the comparisons hinge on identifying probabilities of methylation states, obtained by simulation, (Figs.6 and 7) with “*normalized methylation rates*” (Table 1). Why is this a reasonable comparison? Please provide some justification for it.

(4) Remarks, questions and misprints.

- page 2, para 1, line 3. “*phosphotase*” \rightarrow phosphatase.
- page 2, para 1, PTM coding. PMID 22899623 offers a discussion of this issue from a quantitative perspective more in keeping with the spirit of the present paper.
- page 3, para -2, line -2 and page 15, line 3, perhaps add “respectively”?
- page 3, para 3, line -2, “*Readers who are less mathematically inclined*”. I suggest removing this sentence. It is impossible to understand the results—Eqs.5 and 6, for instance—without getting to grips with the mathematics. Furthermore, I suspect that saying this allows the authors to get themselves off the hook. It would be preferable to write this section in a way which could be understood by anyone who wants to understand the results, such as this reviewer, rather than someone who is steeped in the details of previous work in the field.
- page 4, para 3, line 2, “*a precise sequence*”. Well, strictly speaking, they do not follow “a” sequence: the demethylase has to follow the reverse sequence to the methylase.
- page 4, line before Eq.3, “*only allowing*”. That is not consistent with Eq.3. The rates depend on the average activity, not the activity of an individual receptor. An activated receptor could therefore be methylated and an inactive receptor could therefore be demethylated, at some rate.
- page 5, above Eq.4, “ P_m ”. Perhaps explain that this means probability?
- page 9, line 2, “*clear*” \rightarrow clearly.
- page 11, caption, line -1, “*MeAsp*”. Perhaps explain what this means for those readers who are not familiar with chemotaxis terminology?
- page 12, line -4, “*four methylation sites*”. Readers more familiar with eukaryotic PTM would expect methylations on lysine or arginine. It would help to explain the different bacterial context, in which methylations may be found on glutamate and other atypical residues.
- page 13, Table 1. It would be helpful to provide at least minimal explanation of how these data were acquired. In particular, these are in-vitro measurements, undertaken with tritiated SAM, without a demethylase. It seems that the data for EEEE and EEQE were taken from Table I in citation [31] and the data for EEDE were taken from Table II in citation [30]. Since they are almost the same, it would be less confusing to take them all from the former. Table I in citation [31] gives data for 13 mutants. On what basis were the 3 mutants described in the paper chosen? What can be said by simulation about the other data? If some of them are inconsistent with the model, it is important to make this clear. This would help to define the extent of the model’s validity and may possibly suggest missing features that need to be considered. If the other data are consistent with the model, then, surely, it is more compelling to say so?

- page 13, para 1, line 4, “*enhancer residues*”. I had no idea what these were and it would be helpful to explain more fully what is going on. In particular, the 7 amino acid separation between methylation sites suggests that the residues lie on the same side of the alpha helix and the rate of the methyltransferase appears to be affected by the charge of the residue which is 7 residues towards the C terminus of the site being methylated. The relevant papers, citations [30] and [31], are now over 25 years old and more recent discussions of this interesting mechanism would be helpful. Furthermore, these additional sites are not part of the models considered in the text previously but they seem to be essential for the simulations whose results are being compared to data. It would be helpful to clarify these enhancer sites further. In particular, I did not find the introduction of an enhancer site for the demethylase to be compelling. There is no “*symmetry*” between modification and demodification: the enzymology is entirely different. To what extent is this assumption essential for the simulations which follow? It would be good to show that this is not just a convenient dodge to make the simulations turn out right.
- page 13, title, “*confirms sequential modification*”. I feel this misrepresents the role of the model (PMID 24886484). The data may be consistent with a sequential model but that does not “*confirm*” that the modifications are sequential. A different model might also explain the data (point (1).2 above). The only way to confirm sequentiality is through experiments.
- page 13, line -1, citation [27]. This reference is incomplete: there are two papers by Sourjik and Berg in that volume of PNAS. Fig.3 of the later paper, PNAS 99:12669-74, has a panel b which plots sensitivity to MeAsp. If these are the data being used, please identify them precisely.
- page 15, Fig.6 and page 17, Fig.7. For Fig.6, what value was used for k^+ and how was this chosen? For Fig.7, what values were used for k^+ and k^- and how were these chosen?
- page 17, lines -6 to -4, “*sequential modification ... guarantees the methylation/demethylation rates to be independent of the receptor methylation level*”. I do not understand this. Eq.3 defines the methylation and demethylation rates in terms of the methylation level. In what sense are the rates then “*independent*” of this quantity?
- page 18, para 3, line 2, “*prevailing*”. Perhaps you mean to say “preceding”?

Point-to-point response to reviewer #1

Remarks to the Author

This manuscript reports a model for sequential methylation of bacterial chemoreceptors. It has been appreciated for some time that the high precision of adaptation observed for the chemotaxis system of *E. coli* implies that adaptive methylation/demethylation rates should depend only on receptor activity. While some studies have addressed deviations from precise adaptation and incorporated the limits of adaptation set by the number of modification sites, there has been no clear picture of what mechanism could be responsible for methylation/demethylation rates being (largely) independent of existing methylation levels. The authors make a compelling proposal that sequential methylation could be this underlying mechanism. Importantly, they support their suggestion by careful consideration of previous experimental observations in Refs. 30 and 31: (1) modification rates at a given site depend on the identity of a residue two helix turns away that would be altered during the course of methylation/demethylation, and (2) mutations to these "regulatory" residues led to differential methylation/demethylation rates that are consistent with the sequential modification hypothesis. The authors also explore the implications of sequential modification, and show that the model is consistent with observations of both precision of adaptation and of response gain. Finally, the authors make additional testable predictions. This work addresses a long-standing mystery in the field of bacterial chemotaxis and will therefore likely be of significant interest to the readership of *Nature Communications*. However, there are a number of issues that must first be addressed.

We thank the reviewer for the careful reading of our manuscript and the positive assessment on the novelty and importance of our work. In the following, we address the remaining issues raised by the reviewer. The main changes are marked in blue in the revised manuscript (main text and SM).

Major issue

1. Necessity for a sequential model. While the authors' sequential model is very ap-

pealing, it is not the only possible mechanism that could underly precise adaptation. Any mechanism for which methylation/demethylation rates depend only on receptor activity leads to precise adaptation. E.g. any mechanism in which methylation/demethylation rates are limited by local CheR/CheB activity would behave similarly. I.e. if CheR and CheB bind quickly to available sites, but their catalytic activity is rate limiting, then this would also lead to overall modification rates independent of current methylation levels. As written, the manuscript repeatedly states that sequential modification is the only possible way of achieving precise adaptation, but this is not correct. Given the centrality of this point to the paper, it is essential that the authors acknowledge that there are other ways that precise adaptation might be achieved. Along these lines, the authors need to revise the presentation of their model, which actually only represents a narrow class of possible models.

The reviewer is correct to point out that perfect adaptation can be achieved as long as the methylation rate is independent of the receptor methylation level. The sequential methylation/demethylation mechanism is not the only mechanism to allow such independence, the other mechanism outlined by the reviewer wherein the catalytic rate is rate limiting can also lead to such independence. Within this alternative mechanism, the enzyme (e.g., CheR) binds to all available active sites randomly, thus the number of available sites effectively changes the substrate concentration. Given that the substrate concentration is much higher than the Michaelis-Menten constant, the methylation reaction rate, which is limited by the slow catalytic reaction, would be independent of the substrate concentration and thus independent of the number of available methylation (demethylation) sites.

*However, the enzyme binding rate (the on-rate k_{on}) in this alternative mechanism would depend on the substrate concentration and thus the available modification sites, which seems to be inconsistent with the recent *in vitro* measurements of the k_{on} rates for CheR binding to Tar(EEEE) and Tar(QQQQ) receptors by Li and Hazelbauer (Li and Hazelbauer, *Protein Science*, 29, 443 (2020)), who showed that k_{on} remains roughly unchanged for Tar(EEEE) and Tar(QQQQ). Furthermore, this alternative scenario is not likely the mechanism in *E. coli* Chemotaxis as it would lead to a random pattern of methylation, which is inconsistent with the experimental observations such*

as those reported by the Koshland lab (ref. 34,35&37 in our paper).

Of course, the detailed molecular mechanism of methylation/demethylation process in *E. coli* chemotaxis is still largely unknown (at least quantitatively) as we discussed on page 22-23 in the Summary Section V of our manuscript. For example, we do not know the binding/unbinding rates of enzymes to the receptor or whether these rates depend on the methylation state of the receptor. Moreover, given the tight receptor cluster, we do not know how different receptors compete for the enzymes, which can also methylation/demethylate neighboring receptors in the cluster. Without knowing these details, the model we used in this paper is an approximation of this complex process rather than a full account of the process with all possible scenarios considered. However, we think some of the general conclusions regarding adaptation accuracy, signal gain, the accuracy-gain tradeoff and their dependence on sequentiality of the underlying multisite modification process should hold true.

To summarize, we agree with the reviewer that the necessity of the sequential modification is not warranted from a purely mathematical (or modeling) point of view. We have removed/replaced the word "necessary" in the text. We have also revised our manuscript to discuss this possible alternative mechanism for perfect adaptation in a new paragraph in Section V on page 21-22. In addition, we have included the mathematical details of a model for this alternative mechanism in the SM.

Minor issue

2. In S4, third line "Fig. 2b,c close to each other..." I believe should be "Fig. 2a,b..." and "stepper" → "steeper".

Done.

Point-to-point response to reviewer #2

1. General comments

Let me begin by apologising for the delay in completing this review, which was due to some unexpected personal issues.

This is an interesting paper which addresses two issues of broad interest for understanding cellular information processing. First, what is the functional benefit of having multiple sites of protein post-translational modification (PTM)? Second, given that there are multiple sites on a protein, is there a functional advantage in enzymatic modification and demodification being sequential? The answers offered here are that multisite PTM of receptor proteins enables chemotaxis in *E. coli* to be both sensitive and adaptive and that sequentiality of multisite PTM supports this and may be necessary for it (below). Sequentiality has frequently been assumed in mathematical models, purely for convenience and with little experimental justification, so it is of considerable interest that this paper suggests that it might actually be favoured biologically. Overall, this is an important study which builds on a substantial body of previous work on bacterial chemotaxis by the senior author. However, there are issues that need clarifying, primarily to do with explaining the underlying mathematical model. In this respect, I was unable to work out exactly what the authors were doing and this problem ought to be fixed. I hope the comments below are of some help in doing so.

I believe the review process is improved by transparency and therefore do not wish to remain anonymous.

Jeremy Gunawardena

We thank the reviewer for the positive assessment on the novelty and importance of our work. We are particularly grateful for the reviewer for his careful reading of our paper and the detailed questions regarding the presentation of our model and results. In the following, we address these issues raised by the reviewer and describe the changes we made in accordance with the reviewer's comments and suggestions. The main changes are marked in blue in the revised manuscript (main text and SM).

2. Specific comments

(1) **Model details.** Previous studies have used the sequential model, whose methylation states are shown in Fig.1a. The present paper is noteworthy in considering also the so-called “random” model in Fig.1b. However, I could not find a mathematical treatment of the random model in the main text or the SM. Citation 14 is given as a reference but this review paper appears to only treat the sequential model. The random model has a lot of moving parts: an individual receptor has, in addition to the methylation states shown in Fig.1b, two additional binary state variables, given by whether or not ligand is bound and whether or not the receptor is active and the receptor cluster is a population of interacting receptors. These additional degrees of freedom are averaged over in the present paper through a timescale separation, in which binding, activation and receptor interactions are treated as fast compared to methylation dynamics. This may be reasonable but it means that there is a “black box” whose inner workings are passed off to reference 14 which, as noted above, does not, in fact, work through the details of the random model. I found the lack of mathematical details problematic for several reasons.

We improved, extended, and reorganized the description of the models in section II of the main text. More mathematical details were included in this revised section and in the revised section S2 of the SM.

1. **It is not clear how the key equations (Eq.4) are derived.** Reference is made to the SM but §S1 jumps from the master equation for methylation states only (Eq.S1) to equations for the total methylation level, m , (Eq.S2) without explaining how to get from the former to the latter and without introducing the activities which appear in Eq.4. Moreover, the equations are only stated for $\eta = 0$ (Eq.S2a) and $\eta = 1$ (Eq.S2b), which is strange when the paper discusses the full range $0 \leq \eta \leq 1$. I presume the authors have worked these matters out in full, so I do not understand why they have not provided the details. I appreciate that there are notational headaches to overcome when describing random models of this kind but that is what the SM is for, to avoid frightening readers who are “less mathematically inclined”. For similar kinds of models, we have found that using set notation makes it much easier

to express the mathematical details; see, for instance, PMID 27368104 for binding models and PMID 129945451 for describing PTM patterns.

We have now included more details in the SM on how the key equations are derived, which include the steps from Eq. S2 to Eq. (S4) and Eq. (S9) (previous Eq. S1 to Eq. S2). Regarding Eq. (8) (previous Eq. 4), we included the mathematical definition of the relevant statistical quantities in the main text (Eq. (2) to Eq. (5)). We improved the discussion of these quantities before their definition, and included discussion about the special cases $\eta = 0$, $\eta = 1$, and the more general cases ($0 < \eta < 1$) before and after Eq. (8).

The set notation used in PMID 27368104 is indeed useful to describe the transitions between states of system with multiple binding sites. We used a similar approach when writing the simulation code. However, we believe that with the more detailed descriptions of our model the revised manuscript is clear enough without the need for presenting/using a different set of mathematical notations.

- 2. As a consequence of the previous point, the underlying assumptions are not specified clearly and this makes the conclusions of the paper less compelling.** For instance, it seems to follow from Eqs.2 and 3 that there are only two free parameters for the methylation dynamics, k^+ and k^- , but the forward modification and backward demodification rates are scaled in terms of the quantity $\langle a \rangle_m$. I can see why this might help in deriving Eq.4 but for an enzyme to sense something that depends on the state of the entire receptor cluster seems like a strong assumption. Is the enzyme sampling the cluster in some way? Moreover, is it reasonable to claim that a sequential model is necessary for the conclusions that are drawn when the random model to which it is being compared is so special? With more subtle assumptions, perhaps a random model might also be capable of approximately perfect adaptation? At least, how can this be ruled out?

This is an important point because the necessity for sequentiality is one of the main contributions of the paper. It is a surprising result because the feedbacks are all the same, so why should the presence of non-sequential methylation patterns compromise perfect adaptation? I would like to have a clear reason for this but was unable to disentangle one from the text. I strongly encourage the authors to provide it.

The reviewer is correct in pointing out that “for an enzyme to sense something that depends on the state of the entire receptor cluster seems like a strong assumption.” Indeed, this was not assumed in our model, where the methylation/demethylation rate of a receptor only depends on its own activity (or conformation). The mean activity $\langle a \rangle_m$ is introduced when we analyze the mean methylation/demethylation rate k_m^\pm , which is the mean methylation (demethylation) rate averaged over all the receptors with methylation level m . To clarify this point, in the revised manuscript we have now properly defined $\langle a \rangle_m$ in Eq. (4) before introducing Eq. (6) (previous Eq. 3).

Intuitively, the previously established condition for perfect adaptation is that the methylation (demethylation) rate of a chemoreceptor depends only on its activity. For a receptor with a single modification site, this condition can be satisfied by having the catalytic rate of the enzymatic reaction depends on the activity. However, for a receptor with modification methylation sites such as the bacterial chemoreceptors, there needs to be additional requirements for satisfying this perfect adaptation condition. For example, in the random modification scheme, the methylation rate of the receptor is the catalytic rate of a single active site, which depends on the receptor activity, multiplied by the number of available sites ($M - m$). This explicit dependence of the methylation rate on the methylation level m breaks the perfect adaptation condition. On the other hand, for the sequential modification scheme, the number of available modification site is always 1 (except at $m = M$ when it is zero), so the perfect adaptation condition is satisfied.

As suggested by the reviewer, we have now included a new paragraph right after Eq. 8 to provide an intuitive understanding of the importance of sequential modification for perfect adaptation.

3. **The notation is confusing.** The “state” of the random model sometimes refers to the pattern of methylations on the receptor, $\vec{m} = (m_1, \dots, m_4)$, and sometimes to the total methylation level, $m = \sum_i m_i$, but these symbols are not used consistently. For instance on page 4 (para 4, lines 1 and 2), “For the general case, ... state- m is given by ...”, seems to refer to the random (general) model but its state is described by m

rather than by \vec{m} . I suggest carefully distinguishing these two cases or, preferably, not using the same letter for both \vec{m} and m . The problem becomes acute with Eq.4b. What does a m mean in this formula? It occurs inside the average, so it presumably refers to an individual receptor. But how does a receptor activity, a , depend on a total methylation level m ? If mathematical details had been given, it might have been possible to work this out but I am afraid I am stumped.

We agree with the reviewer that using both \vec{m} and m may cause confusion. To better differentiate similar quantities, we now use $\vec{\mu}$ instead of \vec{m} for representing the methylation state of a given receptor. The component of $\vec{\mu}$, μ_j , with $j = 1, 2, \dots, M$ labeling the modification sites, is a binary number ($\mu_j = 1$ – methylated; $\mu_j = 0$ – unmethylated). The total methylation level is still given by m : $m \equiv \sum_{j=1}^M \mu_j$. We use the index $i = 1, 2, \dots, N$ to represent the individual receptors in the system (receptor cluster) with N receptors. Now, the vector $\vec{m} = (m_1, m_2, \dots, m_N)$ represents the methylation levels of all the receptors in the system. Finally, we introduce the tilde on \tilde{k}_j^\pm in Eq. (7) to denote the site-specific methylation rate.

We believe that the problems regarding Eq. (8) (previous Eq. 4) are resolved by the changes we made, which are described above.

(2) **Gain and error.** The meaning of, and the relationship between, these two properties is hard to grasp. It would help to explain perfect adaptation before defining the adaptation error (Eq.6). If that had been done, it would be easy to justify the error as the difference between the adapted activities at the ends of the range, as on the RHS of Eq.6. It would not be necessary to define it in terms of a logarithmic derivative which is then integrated, as on the LHS of Eq.6. This derivation seems unnecessarily complicated and the integral formula is particularly strange because it looks very much like the formula for the gain given in Eq.5. The relationship would be even clearer if, in Eq.5, the lower and upper limits were written correctly, and, in Eq.6, $\langle a \rangle^A[L]$ was written in place of $\langle a \rangle^A[L]$ and \log_{10} in place of \log . The difference between these two formulas then lies in the terms,

$$\left. \frac{\delta \langle a \rangle}{\delta [L]} \right|_{(m^A[L])} \quad \text{and} \quad \frac{\delta \langle a \rangle^A}{\delta [L]}$$

in Eq.5 and Eq.6, respectively. The latter is straightforward: the average activity after adaptation is a function of the ligand concentration and the derivative is being taken. As

for the former, it is stated that δa is the “immediate (fast) response ... resulting from a small change $\delta[L]$ in the stimulus”. There are several things I do not understand about this.

We now added a general definition of adaptation in Section I on page 2-3.

We have also made significant changes (on page 8) to clarify the definition of the gain, which is the more subtle one as pointed out by the reviewer. See below for our responses to the detailed questions.

1. The meaning of the former quantity now depends on the timescale separation that leads to the “black box” referred to in point (1), whose details have not been given. It appears to be a transient, time-dependent quantity (“fast”) in contrast to the latter quantity, which is a steady-state value. If so, at what time is the value being determined? No time is given in the formula.
2. Why is δ used here, in place of d ? The implication is that it is a finite difference rather than an infinitesimal derivative. But no higher-order terms are present in the formula, so why is it necessary to make this distinction?
3. What does the limitation to $m^A[L]$ mean? Normally, such a limitation specifies the value of the independent variable, which in this case is $[L]$. But a total methylation level cannot be a ligand concentration. Again, I am stumped.

We have now made the definition of the gain more explicitly (page 8). The key point is that the activity depends on both the methylation levels of the receptors, which vary slowly and the ligand concentration. For computing the immediate response $\delta\langle a \rangle$ upon a change of ligand concentration from $[L]$ to $[L] + \delta[L]$, we take the methylation dynamics to be infinitely slow, i.e., the receptor methylation levels remain unchanged at their pre-stimulus levels \vec{m}^A that are adapted to the previous ligand concentration $[L]$. Therefore, we have:

$$\delta\langle a \rangle \approx \langle a \rangle \left(\vec{m}^A([L]), [L] + \delta[L] \right) - \langle a \rangle \left(\vec{m}^A([L]), [L] \right) \approx \left. \frac{\partial \langle a \rangle}{\partial [L]} \right|_{\vec{m}^A([L])} \times \delta[L].$$

The limitation for the partial derivative in the above equation means that the methylation levels of all the receptors in the receptor cluster represented by \vec{m} remain

frozen (fixed) at their pre-stimulus levels $\bar{m}^A([L])$ that are adapted to the previous ligand concentration $[L]$.

Finally, we used $\delta[L]$ instead of the infinitesimal $d[L]$ because in our simulations and in the previous work, e.g., by Sourjik and Berg (PNAS 2002), where the gain was first defined quantitatively, $\delta[L]$ is taken to be a small but finite fraction of $[L]$: $\delta[L] = 0.1[L]$. Another reason that the gain is a partial derivative with respect to $[L]$ while keep \bar{m} fixed and the error depends on the total derivative with respect to a change in $[L]$. We used $\delta[L]$ to distinguish it from the total derivation used in the definition for the error.

We have now included these clarifications described above in the revised text (on page 8).

(3) Experiments and predictions. The comparisons to experimental results are appealing and substantially enhance the paper. However, the comparisons hinge on identifying probabilities of methylation states, obtained by simulation, (Figs.6 and 7) with “normalized methylation rates” (Table 1). Why is this a reasonable comparison? Please provide some justification for it.

We are glad that the reviewer likes the comparison between our modeling results to previous experiments. In the experiments, the methylation rates were not measured directly, they were determined (inferred) by measuring the concentrations of receptors in different methylation states and by assuming the rates are proportional to the concentrations of the end state. Therefore, only the ratios between these measured quantities can be used to compare with those from our model, but not their absolute values. To clarify and justify the comparison, we have added the following sentences in Section IV A when we describe the comparison: “After being methylated with tritiated SAM, the receptors were cleaved and the extent of methylation of each site was determined by high-performance liquid chromatography. The methylation rates were calculated in arbitrary units, reproduced here in Table I. Since the absolute values are not available, we can only analyse the relative methylation ratios of the different sites and mutants.”

(4) Remarks, questions and misprints.

Thank you for bring these problems to our attention. We dealt with them as you suggested.

- page 2, para 1, line 3. “*phosphotase*” → phosphatase.

Changed.

- page 2, para 1, PTM coding. PMID 22899623 offers a discussion of this issue from a quantitative perspective more in keeping with the spirit of the present paper.

Thank you for letting us know about this paper. We included it in the discussion.

- page 3, para -2, line -2 and page 15, line 3, perhaps add “respectively”?

Added.

- page 3, para 3, line -2, “*Readers who are less mathematically inclined*”. I suggest removing this sentence. It is impossible to understand the results — Eqs.5 and 6, for instance — without getting to grips with the mathematics. Furthermore, I suspect that saying this allows the authors to get themselves off the hook. It would be preferable to write this section in a way which could be understood by anyone who wants to understand the results, such as this reviewer, rather than someone who is steeped in the details of previous work in the field.

Removed.

- page 4, para 3, line 2, “*a precise sequence*”. Well, strictly speaking, they do not follow “a” sequence: the demethylase has to follow the reverse sequence to the methylase.

We did the following change: “the methylation and demethylation processes follow the same sequence (in opposite directions)”

- page 4, line before Eq.3, “*only allowing*”. That is not consistent with Eq.3. The rates depend on the average activity, not the activity of an individual receptor. An activated receptor could therefore be methylated and an inactive receptor could therefore be demethylated, at some rate.

The phrase “only allowing methylation (demethylation) of the inactive (active) receptors respectively” (now before Eq. (6)) is correct, and refers to individual receptors. To make this point clearer, we properly distinguish the behavior of individual receptors and of the population before introducing Eq. (6) (previous Eq. (3)).

- page 5, above Eq.4, “ P_m ”. Perhaps explain that this means probability?

This quantity is now properly introduced by Eq. (2).

- page 9, line 2, “clear” → clearly.

Changed.

- page 11, caption, line -1, “MeAsp”. Perhaps explain what this means for those readers who are not familiar with chemotaxis terminology?

We replaced “MAsp” by “methyl aspartate”.

- page 12, line -4, “four methylation sites”. Readers more familiar with eukaryotic PTM would expect methylations on lysine or arginine. It would help to explain the different bacterial context, in which methylations may be found on glutamate and other atypical residues.

We made explicit the, in the first paragraph of Section IV, the peculiarity of methylation of glutamate residues in bacterial chemotaxis.

- page 13, Table 1. It would be helpful to provide at least minimal explanation of how these data were acquired. In particular, these are in-vitro measurements, undertaken with tritiated SAM, without a demethylase. It seems that the data for EEEE and EEQE were taken from Table I in citation [31] and the data for EEDE were taken from Table II in citation [30]. Since they are almost the same, it would be less confusing to take them all from the former. Table I in citation [31] gives data for 13 mutants. On what basis were the 3 mutants described in the paper chosen? What can be said by simulation about the other data? If some of them are inconsistent with the model, it is important to make this clear. This would help to define the

extent of the models validity and may possibly suggest missing features that need to be considered. If the other data are consistent with the model, then, surely, it is more compelling to say so?

Although reference [36] (previous [31]) contains all mutants, it presents the EEDE datum with lower precision than the original information in [38] (previous [30]). We need the full precision values when calculating the parameter η in Section S6 of the SM.

We extended the discussion of these experimental data in the third paragraph of page 18 and in footnote [41], including the reason behind our pick of the data. We included one more mutant $EEE(N)E$ in Table I, and the corresponding discussion in Section IV A.

- page 13, para 1, line 4, “*enhancer residues*”. I had no idea what these were and it would be helpful to explain more fully what is going on. In particular, the 7 amino acid separation between methylation sites suggests that the residues lie on the same side of the alpha helix and the rate of the methyltransferase appears to be affected by the charge of the residue which is 7 residues towards the C terminus of the site being methylated. The relevant papers, citations [30] and [31], are now over 25 years old and more recent discussions of this interesting mechanism would be helpful. Furthermore, these additional sites are not part of the models considered in the text previously but they seem to be essential for the simulations whose results are being compared to data. It would be helpful to clarify these enhancer sites further. In particular, I did not find the introduction of an enhancer site for the demethylase to be compelling. There is no “symmetry” between modification and demodification: the enzymology is entirely different. To what extent is this assumption essential for the simulations which follow? It would be good to show that this is not just a convenient dodge to make the simulations turn out right.

We changed from the word “enhancer” to the word “initiator” to clarify its function in the model. We include a definition of initiator residues after Eq. (7) and discussion on second paragraph of Section IV. We used the newly included data for $EEE(N)E$ in Table I to discuss the role played by the initiator h_{3f} in the

experimental results of Section IV A.

Unfortunately, cheB methyltransferase activity has not been investigated in details, and we could not provide recent experimental support on this basis, favorable or against our assumptions regarding CheB behaviour. Notwithstanding, we present in Section S9 an indirect experimental evidence based on an experimental work from Nowlin et al [43, 44] that investigate the deamidation activity of cheB. In the revised manuscript, we rewrote the discussion about sequential behavior of cheB on the last paragraph of Section IV without references to symmetry and reproduced an argument connecting cheB on to its structural similarities with CheR from a work by Djordjevic et al [39].

So far, there has not been many experimental papers on methylation sequentiality. Besides from the previous work from the Koshland lab, which we studied in detail in our paper, we also discussed a work from 2004 by Perez et al [37] that is relevant. In this work, not only the receptor methylation sites were altered, but also relevant residues of cheR. Their results support the hypothesis that the interaction of cheR with a residue in the receptor seven residues away from the methylation site is important for methylation, but there is no quantitative results in the Perez paper that can be compared with our theory directly.

- page 13, title, “*confirms sequential modification*”. I feel this misrepresents the role of the model (PMID 24886484). The data may be consistent with a sequential model but that does not “confirm” that the modifications are sequential. A different model might also explain the data (point (1).2 above). The only way to confirm sequentiality is through experiments.

We changed from “confirms” to “supports”.

- page 13, line -1, citation [27]. This reference is incomplete: there are two papers by Sourjik and Berg in that volume of PNAS. Fig.3 of the later paper, PNAS 99:12669-74, has a panel b which plots sensitivity to MeAsp. If these are the data being used, please identify them precisely.

We double checked the reference and it is correct. In the later PNAS paper by Sourjik and Berg entitled “Binding of the Escherichia coli response regulator

CheY to its target measured in vivo by fluorescence resonance energy transfer” (PNAS 99:12669-74, 2002), the authors studied the binding of CheY-P to FliM, which is outside the scope of our study here.

- page 15, Fig.6 and page 17, Fig.7. For Fig.6, what value was used for k^+ and how was this chosen? For Fig.7, what values were used for k^+ and k and how were these chosen?

We included the required information and explained its origin in the caption of Fig. 7: “Parameters used here are $\eta = 0.1$, $[L] = 100 \mu\text{M}$, $k^+ = 0.7 \text{ s}^{-1}$, and $k^- = 1.4 \text{ s}^{-1}$. The values of k^+ and k^- are observed at the black diamond in Fig. 4.”

- page 17, lines -6 to -4, “sequential modification . . . guarantees the methylation/demethylation rates to be independent of the receptor methylation level”. I do not understand this. Eq.3 defines the methylation and demethylation rates in terms of the methylation level. In what sense are the rates then “independent” of this quantity?

We have now made explicit, in the second paragraph of Section V, the reference to Eq. (8a) to clarify the meaning of that statement.

- page 18, para 3, line 2, “prevailing”. Perhaps you mean to say “preceding”?

Indeed. Changed.

REVIEWERS' COMMENTS:

Reviewer #1 (Remarks to the Author):

The authors have satisfactorily addressed my concerns, and I am happy to recommend acceptance of the manuscript in its current form.

Reviewer #2 (Remarks to the Author):

Sequential modification of bacterial chemoreceptors is key for achieving accurate adaptation and high gain simultaneously

Mello, Beserra and Tu

Nature Communications paper # NCOMMS-19-27808A (revised version)

General comments

The authors have made a careful and conscientious attempt to address the issues which I raised previously and to explain what they have done clearly, for which I commend them. I found the changes and explanations helpful. The one issue on which I feel the authors have not offered the best response has to do with point (1), regarding the details of the model. The full model is still not formally described, either in the main text or the SI, rather a more complete analysis is given for the extreme cases $\eta = 0$ and $\eta = 1$. What was not clear to me previously is that the intermediate cases, $0 < \eta < 1$, are dealt with by simulation, rather than analytically. The authors may feel that this lessens the value of writing down the full model in formal mathematical language. I appreciate that this is not straightforward, as it does indeed require “*a different set of mathematical notations*” (response, page 6), but I feel there is an obligation to do so despite the inconvenience involved. After all, the authors are asking us to rely on their simulation code, which is a much less formal description of the model and one which is not, so far as I can see, publicly available. I do not think it is fair to the authors to press this point further, and I am happy to recommend publication of the paper, but I would encourage the authors to make their simulation code accessible to readers.

Jeremy Gunawardena

RESPONSE TO REVIEWER 2

The authors have made a careful and conscientious attempt to address the issues which I raised previously and to explain what they have done clearly, for which I commend them. I found the changes and explanations helpful. The one issue on which I feel the authors have not offered the best response has to do with point (1), regarding the details of the model. The full model is still not formally described, either in the main text or the SI, rather a more complete analysis is given for the extreme cases $\eta = 0$ and $\eta = 1$. What was not clear to me previously is that the intermediate cases, $0 < \eta < 1$, are dealt with by simulation, rather than analytically. The authors may feel that this lessens the value of writing down the full model in formal mathematical language. I appreciate that this is not straightforward, as it does indeed require “a different set of mathematical notations” (response, page 6), but I feel there is an obligation to do so despite the inconvenience involved. After all, the authors are asking us to rely on their simulation code, which is a much less formal description of the model and one which is not, so far as I can see, publicly available. I do not think it is fair to the authors to press this point further, and I am happy to recommend publication of the paper, but I would encourage the authors to make their simulation code accessible to readers.

Response: We made the code available online, as described in the Code Availability section on the main text. We included a pseudo-code of the algorithm to provide a step-by-step description of the Monte-Carlo simulations in the Supplementary method.